# Ergonomic Assessment of Physical Load in Slovak Industry Using Wearable Technologies

**Daniela Onofrejova ***[iD]**, Michaela Balazikova, Juraj Glatz, Zuzana Kotianova** [iD] **and Katarina Vaskovicova**

Department of Safety and Production Quality, Institute of Special Engineering Processologies, Faculty of Mechanical Engineering, Technical University of Kosice, Letna 1/9, 042 00 Kosice, Slovakia; michaela.balazikova@tuke.sk (M.B.); juraj.glatz@tuke.sk (J.G.); zuzana.kotianova@tuke.sk (Z.K.); katarina.vaskovicova@student.tuke.sk (K.V.)

**\*** Correspondence: daniela.onofrejova@tuke.sk; Tel.: +421-55-602-2513

**Abstract:** The physical tasks of workers are demanding, particularly when performed long-term in unsuitable working position, with high frequency, heavy load, after injury, with developing damage of health or reduced performance due to advanced age. Work-related musculoskeletal disorders (WMSDs) result from overuse or develop over time. Work activities, which are frequent and repetitive, or activities with awkward postures, cause disorders that may be painful during work or at rest. There is a new technology in the market, occupational exoskeletons, which have the prerequisites for minimizing the negative consequences of workload on WMSDs. We provided pilot quantitative measurements of the ergonomic risk at one selected workplace in a Slovak automotive company with four different workers to prove our methodology using wearable wireless multi-sensor systems Captiv and Actigraph. At first, the test was performed in standard conditions without an exoskeleton. The unacceptable physical load was identified in considerable evaluated body areas—neck, hip, and shoulder. Next, the passive chair exoskeleton Chairless Chair 2.0 was used in trials as an ergonomic measure. Our intention was to determine whether an exoskeleton would be an effective tool for optimizing the workload in selected workplaces and whether the proposed unique quantitative measurement system would give reliable and quick results.

**Keywords:** ergonomic risk assessment; physical load; exoskeleton Chairless Chair 2.0; human health prevention; work-related musculoskeletal disorders



## 1. Introduction

Work-related musculoskeletal disorders (WMSDs) are a group of painful disorders of the muscles, tendons, and nerves. The European Agency for Safety and Health at Work [1] includes acute traumas and fractures within the WMSDs group, which means traumatic injuries to the muscles, tendons, and nerves due to accidents. Frequent and repetitive work activities, or activities with awkward postures cause disorders that may be painful during work or at rest.

Damage to the musculoskeletal system is one of the most common work-related disorders. Recent research indicates that work-related musculoskeletal disorders (WMSDs) are one of the major health problems in the workplace and have a significant economic impact [2,3]. WMSDs affect millions of employees across Europe and represent a cost in billions of euros for employers. Dealing with musculoskeletal disorders helps to improve the lives of workers, but it also makes business perspective.

The innovation potential in digitalization to meet growing demand and increase productivity ranges from increasingly sophisticated robots replacing workers in customer-oriented roles to additive manufacturing technologies (3D printing) producing human organs [4]. New body-worn assistive devices–occupational exoskeletons [5] have been introduced in some workplaces to help workers perform manual manipulation tasks

while reducing the load on the muscular system [6]. Currently, the interest in exoskeleton research has expanded into several areas. In particular, it has recently transferred from the medical/rehabilitation field to the industrial sector. There are several reasons for this. On the one hand, the development of rehabilitation exoskeletons could reach a plateau because reliable and efficient solutions are available for these applications. On the other hand, Industry 4.0 is moving towards the concept of smart factories. The adoption of automation in industry has been growing over the last twenty years, intending to increase productivity while reducing the physical workload required for human workers [7]. Also, according to contributions [8,9], the implementation of robotics and exoskeletons could also contribute to the improvement of working conditions. Pons et al. [10] describe that the topic of exoskeletons is widely presented, including biomechatronic design, cognitive and physical human-robot interactions, wearable robotic technologies, kinematics, dynamics, and control.

Upper- and lower-limb wearable exoskeletons, which are mechanical structures worn on the body to enhance the strength of the wearer, have been developed and studied for their potential effect to limit exposure to physical load [11]. Moreover, kinematics, postural control, and discomfort in passive, lower-limb exoskeleton were studied in [12]. Types of exoskeletons can be classified according to five criteria, which are: 1. what part of the human body the exoskeleton is designed for; 2. what element the exoskeleton is driven by; 3. how the exoskeleton is fixed; 4. how the exoskeleton is controlled; 5. what the exoskeleton is composed of [13].

Currently, most studies on exoskeletons demonstrate promising results. Maurice, J. et al. [14] investigated the passive exoskeleton PAEXO for overhead work, which effectively reduces physical effort and fatigue. Veslin, E.Y. et al. focused on the study of the upper arm exoskeleton and created a simulation in Matlab$^{®}$ [15]. Another study indicated that lower [16] extremity exoskeletons, aiming to reduce the physical load associated with prolonged standing, may impair workers' postural control, and increase the risk of falling. According to Zampogna et al. [17], wearable technology [18–21] has been proving convincing and useful results in evaluating motor impairments of subjects suffering from (among others) Parkinson's disease. Topalidis et al. [22] and Guediri et al. [23] compared performance, reliability, and the absolute error rate of worn ActiGraph GT3X in free-living conditions in young and older adults when measuring physical activity. Strath et al. [24] used physiological and accelerometer data to improve physical activity assessment. Other studies argue that exoskeletons need to be closely linked to the manufacturing activities of Industry 4.0 organizations as they will perform operations in collaboration with these advanced technologies [25,26]. Authors [27] in the study examined the opinion of factory workers and non-workers on three human-centered technologies aiming at improving working conditions: collaborative robots, exoskeletons, and wearable sensors. Workers and non-workers were mostly positive about these technologies and agreed they would increase workers' physical well-being. Authors argue that ethical recommendations must necessarily be complemented by an analysis of the social impact of these technologies, as guidelines for the ethically aligned design of autonomous and intelligent systems do exist. Some studies have investigated poor mental well-being in the workplace due to work-related musculoskeletal disorders [28]. In the automotive industry, the Noonee chairless-chair was investigated, which is a passive device for workers that requires no power. It is supposed to be a practical device for workers who must remain in ergonomically uncomfortable positions [29].

Throughout the European Union, musculoskeletal disorders (MSDs) are the biggest cause of absenteeism, accounting for 40% of workers' compensation costs and a reduction of around 1.6% of gross domestic product [30]. In the US, similar statistics show that MSDs represent 33% of all staff compensation costs [31]. WMSD cases increased from 293 cases in 2019 to 328 cases in 2020. The Accommodation and Food Services industry was the top contributor, accounting for 16% (54 cases) of all WMSD cases, followed by the Manufacturing and Health Services industries with 49 (15%) and 45 (14%) WMSD cases, respectively [32].

European directives [33–38], EU health and safety strategies, Member States' regulations, and best practice guidelines already recognize the importance of preventing musculoskeletal disorders. Risks of damage to the musculoskeletal system related to work fall within the scope of the framework directive on occupational safety and health 89/391/EEC [33] (Act of the Slovak Republic No. 124/2006 Coll. [34]), which aims to protect employees from work-related risks in general and to establish the employer's responsibility for ensuring safety and health at work. The directive requires a risk assessment in the work environment. Identifying risk factors highlight some of the problems faced by employees and the importance of understanding corporate practices to prevent damage to the musculoskeletal system, including the responsibilities of both employers and employees. Article six of the Framework Directive promotes an ergonomic approach, as it requires the employer to adapt work to the individual, in particular, by reducing monotonous work and work at a predetermined pace and reducing the health effects of work.

The aim of this contribution is the quantitative evaluation of physical workload to evaluate the ergonomic risk of using wearable systems. Results from measurement one, labeled as STAND (ergonomic evaluation of the workload at the assembly workplace in standing position), are compared with results from measurement two, labeled as EXO (ergonomic evaluation of the workload at the assembly workplace while applying ergonomic measure—exoskeleton enabling easy change between standing and sitting positions). The objective quantitative measurement will be used to determine the suitability of the exoskeleton for work activity in order to minimize the ergonomic risk as a prevention of worker's health and maintenance of the maximum efficiency and productivity of employees. We assumed, that lower body exoskeleton Chairless Chair 2.0 (CC 2.0) could help improve the body posture of the worker and lower the risk of developing WMSDs.

## 2. Materials and Methods

A wide range of methods and tools (methods such as, e.g., RULA, REBA, NIOSH, EAWS, and software products such as Tecnomatix Jack [9], Captiv [39–42], CERAA [39], XSens Motion Capture [40]) are used in ergonomics for the identification and evaluation of ergonomic hazards. These methods enable us to study, analyze, and evaluate human behavioral patterns while performing a work activity. The combination of these methods with knowledge of biomechanics, the anatomical structure of the body, and the way it responds to a load allows the design of efficient and healthy workplaces. The analysis and evaluation of the ergonomic conditions in a system consider a set of criteria that the system must meet with regard to the standard requirement of adapting the technical elements and working conditions to the performance capabilities of the company employees [7,9].

### 2.1. Measurement Design

Our field experiment is based on comparing the occupational conditions of workers in two states: (a) standard work in a standing position [STAND]; (b) work with a technical device—exoskeleton in a standing/sitting position [EXO]. The introductory measurements were set in an industrial workspace, where the worker's job was to assemble the mounting nest by placing the outer and inner ring there and inserting a metal stone into the marked holes in a takt time of 0.94 min. Synchro pre-assembly consisted of 10 tasks, with each task handling one piece; the mean manipulation time ($TM_{mean}$) for manipulation per one piece was ~0.09 min. A particular assembly workplace with repetitive movements was chosen for exoskeleton deployment. The working height was solid with an "elbow-floor" distance $d_{ef}$ equal to 1.22 m and a working distance "grasping arm" (sagittal plane) $d_{ga}$ equal to 0.23 m. Workplace design could not be changed during the experiment, and we intended to investigate whether the existing workplace was suitable for a diverse group of workers, if there were any insufficiencies, and if the workplace posed a risk of developing MSDs to the employees. The workers as end-users were trained by super-users with safety instructions on using the exoskeleton CC 2.0. The training was performed a day before experiments were conducted and lasted ~30 min for each tested worker. The duration of

one experiment was ~30 min. Environmental measurement conditions were: temperature 22.6 °C, humidity 46.4%, pressure 102 kPa, CO2 level 403.7 ppm. As a corrective measure, the industrial exoskeleton CC 2.0 was applied as a support for lower body parts [16].

The Chairless Chair® 2.0 (CC 2.0) (Figure 1) is a Wearable Ergonomic Mechanical Device intended for use in production and assembly lines. It allows users to take breaks and sit down occasionally while working. The occupational exoskeleton was used in Experiment 2 (EXO) as a technical aid for improving the ergonomic postures of the worker and applying a sit-stand pattern at work. Experiment 1 (STAND) preceded Experiment 2 (EXO) and served to measure the actual state of the ergonomic workload of the worker.

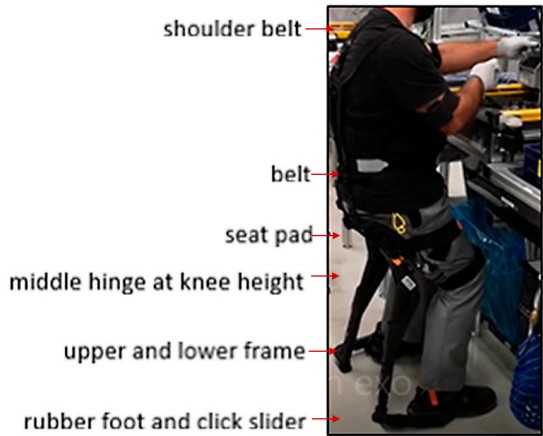

**Figure 1.** Lower body exoskeleton—Chairless Chair 2.0 (CC 2.0). Exoskeleton was used in Experiment 2 (EXO): minimizing the physical load by ergonomic sit-stand pattern.

## 2.2. Study Population

The trial field study participant group consisted of 3 males and 1 female (all right-handed). The age group was characterized as middle-aged adults and old adults (60+) in productive age (Table 1). Workers in a middle-aged group did not report any health disorders. Older adults reported both knees and right hip pain.

**Table 1.** Anthropometric parameters of participated subjects consisted of 3 males and females, age group 40–49, 50–59, and 60+. BMI—Body Mass Index is a measure of body fat based on height and weight that applies to adult men and women.

| N | Gender | Age | Weight [kg] | Height [m] | BMI |
|---|---|---|---|---|---|
| Participant 1 | M | 43 | 81 | 1.68 | 28.70 |
| Participant 2 | M | 59 | 90 | 1.85 | 26.30 |
| Participant 3 | M | 60 | 82 | 1.78 | 25.88 |
| Participant 4 | F | 44 | 85 | 1.68 | 30.12 |

The study was conducted according to the guidelines of the Declaration of Helsinki and approved by the Institutional Review Board of the Technical University of Kosice (protocol code 8268/2021/R-OLP).

## 2.3. Measurement Tools

To assess the ergonomic risk, the wireless sensor ergonomic system TEA Captiv was used simultaneously with another wearable system—smartwatch Actigraph. Captiv was chosen due to its strong features and availability of a multimodal and complex human motion analysis. Steinebach et al. [42] compared the accuracy of motion capture for complex movements using the Captiv system with Microsoft Kinect V2 and considered Captiv preferable for ergonomic analyses in terms of accuracy in the majority of cases. This was especially the case in industrial work environments with occlusions. Peeters et al. 2019 [43] demonstrated the accuracy, reliability of IMU (inertial measurement units) for outdoor

motion capturing in diverse activities (regular tasks such as walking and fast, complicated tasks such as rehabilitation and sports).

Captiv enables an adaptable and scalable solution for capturing workers in their work environments thanks to a multifunctional analysis embodying body posture, carrying capacity, musculoskeletal limitations, and repetitive movements and vibrations. It is a flexible, scalable measurement and analysis toolkit for ergonomics, workplace analysis, occupational safety, HMI (Human-Machine Interface), prototyping, research, VR (Virtual Reality), and more applications [39]. Simple software enables a quick analysis of tasks based on a video, up to a complex multimodal measuring system with synchronous video record. Wireless motion sensors use inertial measurement units (IMU) and 16-channel T-Rec wireless receivers with a range of 15 m in real-time.

Measured joints, in accordance with thresholds for appropriate range of motion in our experiments were (Table 2): Neck, Back (axis: pelvis—vertebral segment T2), Left shoulder, Right shoulder, Left hip (axis: pelvis—upper left leg), and Right hip (axis: pelvis—upper right leg). For correct 3D visualization, the subject was equipped with a sensor on his back (for tracking the upper body) and on his hip (for tracking the lower body).

**Table 2.** Angle thresholds [°] set for monitoring the motion of the measured joints: Neck, Lower Back, Right/Left Shoulder, Right/Left Hip. Thresholds were set according to the author's [41] recommendation and are part of the collected data evaluation and sorting according to the length of time spent in a given position.

| Threshold/Joint | Neck | | Lower Back | | Shoulder | | Hip | |
|---|---|---|---|---|---|---|---|---|
| Unappropriate Area | Orange | Red | Orange | Red | Orange | Red | Orange | Red |
| Flexion/Extension | 15/−10 | 30/−20 | 30/−10 | 45/−20 | | | | |
| Lateral Right/Left Flexion | 10/−10 | 20/−20 | 10/−10 | 20/−20 | | | | |
| Left/Right Rotation | −15/15 | −30/30 | −15/15 | −30/30 | | | | |
| Right/Left Vertical Rotation | | | | | 60 | 90 | | |
| Right/Left Horizontal Internal/External Rotation | | | | | −70/10 | −90/30 | | |
| Internal/External Rotation | | | | | −40/20 | −60/45 | 10/−10 | 30/−20 |
| Right/Left Flexion/Extension | | | | | | | 70/−10 | 100/−20 |
| Right/Left Abduction/Adduction | | | | | | | 20/−10 | 30/−20 |

The measured data were displayed via a 3D avatar (virtual human mannequin), which offered animations of the provided task with visualizations of system evaluation results by marking body segments with green/orange/red colors, indicating fully customizable threshold values for reference angles. Green color means suitable conditions for a segment loading; orange indicates a change in activity that has to be considered, and red indicates inappropriate activity that needs an immediate correction.

Figure 2a shows the placement of 7 motion sensors (MO) on the following segments: Head (forehead), Back (spine on T2), Pelvis, Left and right arms (humerus), Left and right forearms (radius, cubitus), and Upper left and right legs (femur). The Captiv's avatar represents worker activity, and his simultaneous joint angles indicate with color which threshold values exceeded in the monitored joints, see Figure 2a,b and Figure 3a,b.

Measuring multiple biological signals using different types of sensors can significantly improve the accuracy of physical activity parameter estimates, as opposed to measuring only one signal. Monitoring by multiple sensors can be defined as a method based on three or more types of sensors, e.g., skin temperature, ambient body temperature, heat flux, galvanic reaction of the skin, accelerometer, gyroscope, magnetometer, pressure sensor, breathing, etc. More complex activity parameters can be obtained by multimodal sensors measuring energy expenditure, intensity, frequency, sleep duration, number of steps, distance, and speed [24]. To complete such comprehensive data collection, the system Actigraph ActiLife was added to our testing methodology to record, process, and evaluate the data including:

- Carrying time—for the purpose of selecting and evaluating only the time using the device, as it allows excluding the time of not wearing the device.
- Energy expenditure—an estimate of energy expenditure during physical activity in kcal with 5 equations.
- Metabolic rate—allowing determination of the average hourly, daily rate of metabolic transformation using 12 different metabolic algorithms.
- Inclinometer—detects whether the subject is standing, sitting, or lying down and whether the ActiGraph device has been disconnected.
- Detailed Actigraph system data evaluation is not a part of this paper, we concentrated on the Captiv data in our first trials.

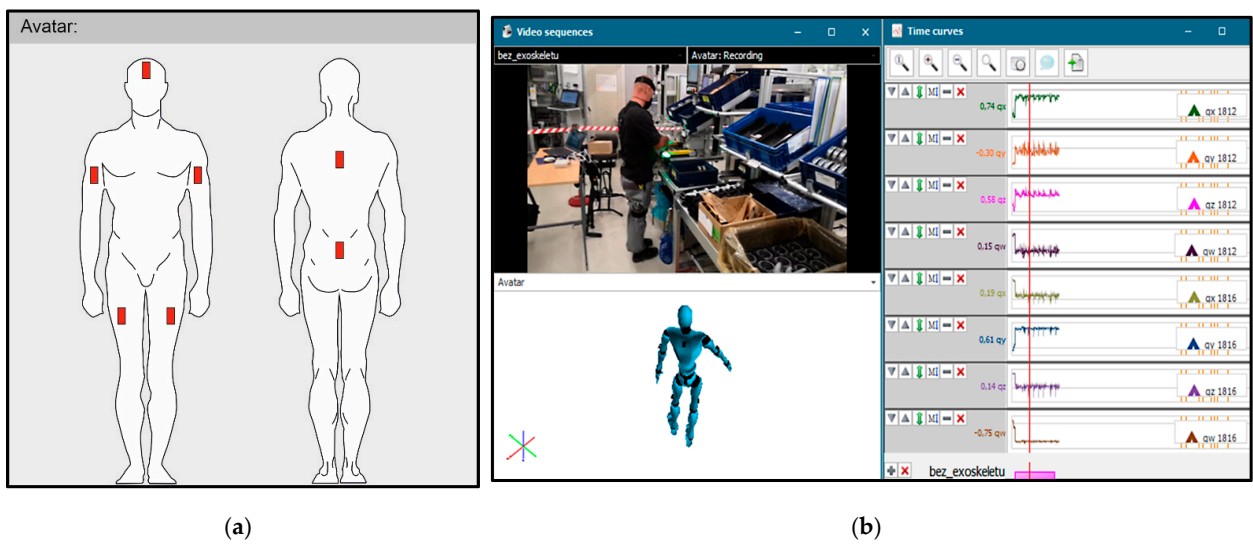

(**a**)　　　　　　　　　　　　　　　　　　　　　　　　　　(**b**)

**Figure 2.** The setting and implementation of the measurement in the industrial workspace: (**a**) Placement of the Captiv wireless sensors on the body for experiments with and without exoskeleton; (**b**) The captured data with Captiv sensor system before their evaluation, with synchronization of data and video recording, and avatar visualization.

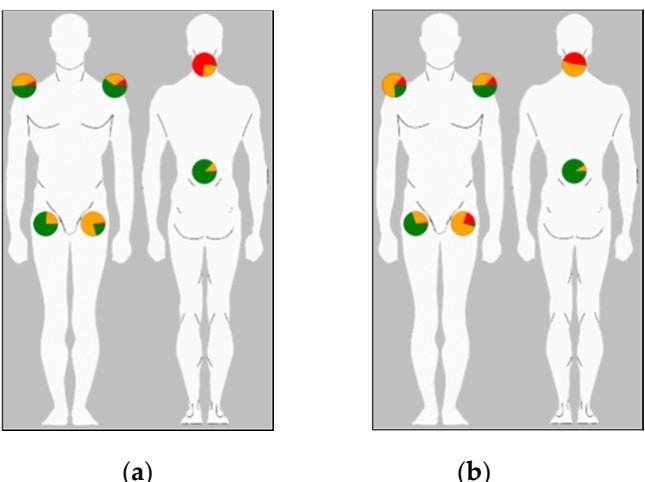

(**a**)　　　　　　　　　　　　　(**b**)

**Figure 3.** Postures. Evaluation as: (**a**) Postures evaluation results for work activity STAND (without exoskeleton); (**b**) Postures evaluation results for work activity measurement EXO (with exoskeleton).

### 2.4. Measurement Method

Our objective was to analyze the physical workload by monitoring the standing working posture of a person in an actual automotive industry workplace, evaluating the length of remaining individual body segments in unsuitable working positions, which in long-term period result in a high level of risk for developing WMSDs. It has already been

confirmed that prolonged standing at work (primarily in one place) increases the risk of low back pain and cardiovascular problems [44–46]. We intended to determine if using the lower body exoskeleton CC 2.0 in the selected standing workplace with repetitive tasks can lower ergonomic risk. The initial experiments were carried out consequently in the order as follows, and this is also procedure for future experiments:

a) Measurement for identification the level of ergonomic risk to the musculoskeletal system;
b) Proposal of corrective measure: use of protective aids, exoskeletons, to determine their suitability in a particular work activity, and next the duration and frequency of its application to workers.

Both above mentioned measuring systems were used simultaneously for ~30 min; observation was interrupted at the end of the working cycle:

c) Captiv—motion analysis, measuring the position/angles of the joints.
d) Actigraph smart watch—metabolic rate, energy expenditure, number of steps, inclinometer.

The Captiv system uses inertial sensors for its angle measurement system [43], in which limb angles are calculated by the integral of angular velocity by the Kalman filter [47]. The angle calculation is a quaternion-based method for 3D measurement of angles in the sagittal, frontal, and transversal planes during the motion of monitored subjects. In the evaluation of motor functions, segment inclination angles and joint angles have important information for researchers. Therefore, studies [48,49] have been performed on the measurement of joint angles or segment tilt angles with inertial sensors. The initialization procedure for Captiv T-Sens Motion sensors calculates the cumulated orientation differences between all sensors while all sensors must be placed in an environment devoiding of magnetic perturbations (no metal, no magnets). Sensors can only be used for measurement after a positive test.

*2.5. Statistical Analysis*

Basic statistical methods and testing of statistical hypotheses were used to analyze the data set measured by Captiv motion sensors in two conditions: worker without exoskeleton CC 2.0 (STAND) and worker wearing CC 2.0 (EXO). To assess normality, we used the Shapiro-Wilk normality test, the most used normality test in the case of a small to medium range up to 2000. When comparing two dependent sets, a paired *t*-test is used, and eventually, a nonparametric Wilcoxon paired test.

When testing statistical hypotheses, the decision to reject or accept the null hypothesis is made using the *p*-value. If the *p*-value is less than the specified level of significance $\alpha$, then the null hypothesis is rejected in favor of the alternative hypothesis. If the *p*-value is equal to or greater than the selected significance level $\alpha$, then the null hypothesis is not rejected. For data assessment, the statistical software R was used. R is an official part of the Free Software Foundation's GNU project and can be downloaded as open-source software for statistical calculations.

**3. Results**

The result of the ergonomic evaluation is a comparison of the measured values with limit values given by the legislation or individually adjustable criteria, classification of works into categories, recommendations for the implementation of measures to reduce physical activity (organizational, ergonomic, technical, etc.). Ergonomic assessments are complementary in accordance with the Council Directives [33–38].

The results in Table 3 represent data for one evaluated worker proceeded by the Captiv system. It describes the ratio of time duration and how long the particular body segment spent in a certain threshold area. The difference method was chosen for comparing the results from both types of experiments, STAND and EXO, to judge quantitative change to find out the level of difference between them; formula 1was used:

$$\Delta E_A = E_{W_A} - E_{N_A} \tag{1}$$

where $\Delta E_A$ is the difference value to compare both types of experiments, $E_{W_A}$ is the value representing the ratio between durations of times when the observable occurred in one of three zones (green + orange + red = 100%) during experiment EXO (with exoskeleton), and $E_{N_A}$ is the value representing the ratio between durations of time when the observable occurred in one of three zones (green + orange + red = 100%) during experiment STAND (without exoskeleton). A change of the ratio $\Delta E_A$ gives difference values, decrease (−) or increase (+). An increase in the green zone means positive change; an increase in the orange zone means positive change only if the red was decreased, and negative change if the green was decreased. An increase in the red zone is always negative.

**Table 3.** Results from the comparative motion measurements with the Captiv system: for one worker when working without exoskeleton STAND ($E_{N_A}$), and with exoskeleton EXO ($E_{W_A}$). The results obtained correspond to the level of exposure to the risk divided in three zones—green, orange, red. $\Delta E_A$—is the difference value to compare both types of experiments, $E_{W_A}$—is the ratio between durations of times when the observable occurs in one of three risk zones for measurement EXO, $E_{N_A}$—is the ratio between durations of times when the observable occurs in one of three risk zones for measurement STAND. The signs +/− in the next column declare positive (+) or negative (−) influence of a particular change in the risk.

| Joint | Movement | Green Area [%] | | | | Orange Area [%] | | | | Red Area [%] | | | |
|---|---|---|---|---|---|---|---|---|---|---|---|---|---|
| | | $E_{NA}$ | $E_{WA}$ | $\Delta E$ | +/− | $E_{NA}$ | $E_{WA}$ | $\Delta E$ | +/− | $E_{NA}$ | $E_{WA}$ | $\Delta E$ | +/− |
| neck | flexion/extension | 10.0 | 19.7 | 9.7 | + | 26.8 | 51.2 | 24.4 | + | 63.2 | 29.1 | −34.1 | + |
| | lateral flexion/extension | 54.6 | 54.9 | 0.3 | + | 44.3 | 40 | −4.3 | − | 1.1 | 5.1 | 4 | − |
| | rotation right/left | 56.2 | 44.4 | −11.8 | − | 32.2 | 42.2 | 10 | − | 11.5 | 13.5 | 2 | − |
| lower back | flexion/extension | 100 | 97.8 | −2.2 | − | 0 | 2.2 | 2.2 | − | 0 | 0 | 0 | + |
| | lateral flexion/extension | 97.4 | 95.4 | −2 | − | 2.6 | 4.1 | 1.5 | − | 0 | 0.5 | 0.5 | − |
| | rotation right/left | 89.5 | 95.3 | 5.8 | + | 9.3 | 4.7 | −4.6 | + | 1.2 | 0 | −1.2 | + |
| right shoulder | rotation external/internal | 87.0 | 79.0 | −8 | − | 11.7 | 16.5 | 4.8 | − | 1.3 | 4.5 | 3.2 | − |
| | vertical rotation | 90.1 | 78.7 | −11.4 | − | 9.6 | 20.4 | 10.8 | − | 0.3 | 0.9 | 0.6 | − |
| | horizontal rotation | 60.6 | 50.0 | −10.6 | − | 33.6 | 42.3 | 8.7 | − | 5.8 | 7.7 | 1.9 | − |
| left shoulder | rotation external/internal | 85.6 | 78.6 | −7 | − | 11.7 | 17.4 | 5.7 | − | 2.7 | 4.0 | 1.3 | − |
| | vertical rotation | 85.5 | 68,6 | −16.9 | − | 11.3 | 23.6 | 12.3 | − | 3.2 | 7.8 | 4.6 | − |
| | horizontal rotation | 67.5 | 69.9 | 2.4 | + | 25.9 | 25.4 | −0.5 | + | 6.6 | 4.7 | −1.9 | + |
| right hip | flexion/extension | 100 | 100 | 0 | + | 0 | 0 | 0 | + | 0 | 0 | 0 | + |
| | abduction/adduction | 99.1 | 97.8 | −1.3 | − | 0.9 | 2.2 | 1,3 | − | 0 | 0 | 0 | + |
| | rotation external/internal | 77.4 | 70.3 | −7.1 | − | 22.5 | 27.5 | 5 | − | 0.1 | 2.2 | 2.1 | − |
| left hip | flexion/extension | 100 | 92.6 | −7.4 | − | 0 | 7.4 | 7.4 | − | 0 | 0 | 0 | + |
| | abduction/adduction | 100 | 99.7 | −0.3 | − | 0 | 0.3 | 0.3 | − | 0 | 0 | 0 | + |
| | rotation external/internal | 20.6 | 10.5 | −10.1 | − | 77.1 | 69.9 | −7.2 | + | 2.3 | 19.6 | 17.3 | − |

The obtained results correspond to the periods when the subject's exposure to the risk was greater than the acceptable threshold. Figure 3 shows the level of activity in the colors (green: appropriate, orange: indicates a change in activity, red: inappropriate). The results in Figure 3a show STAND experiments, and Figure 3b shows the EXO experiments. Comparing the results in Table 3 in the neck joint, there was a visible improvement in neck flexion/extension of 34.1%, the angle exceeding 30° in neck flexion was reduced to the level between 15° to 30°, in neck extension from above −20° to the range of −10° to −20°. However, neck rotation was with or without exoskeleton still in the red area over 10% from the total time and even raised in 2% together with orange raised in ~10%.

Similarly, the risk was raised in the EXO experiments for both shoulders, moving about 10% from the green zone to orange and red. Therefore, we need to improve the height adjustment of the workplace table because, during the sitting position, the situation was worse for shoulder manipulation. The measured data (Figure 3) were displayed via 3D Captiv avatar, in green/orange/red colors based on customizable threshold values for the reference angles. Green indicates suitable parameters, orange indicates a change in activity, and red indicates inappropriate activity with an immediate need for correction.

Results shown in Figure 4 reflect detailed Neck flexion/extension time duration in individual postures in experiments with the exoskeleton. When we look at the hip, the left side was pulled more than the right side, and the left hip rotation in the experiment without the exoskeleton was in the orange color region, representing 77.1% of the total

working time, with the exoskeleton 69.9% time in orange and raised from 2.3% to 19.6% in the red zone; uncertainty and mistrust of a person in the equipment are visible here. A kind of training might be necessary to improve the worker's movement to the right table to pick up a part and then insert the assembled product.

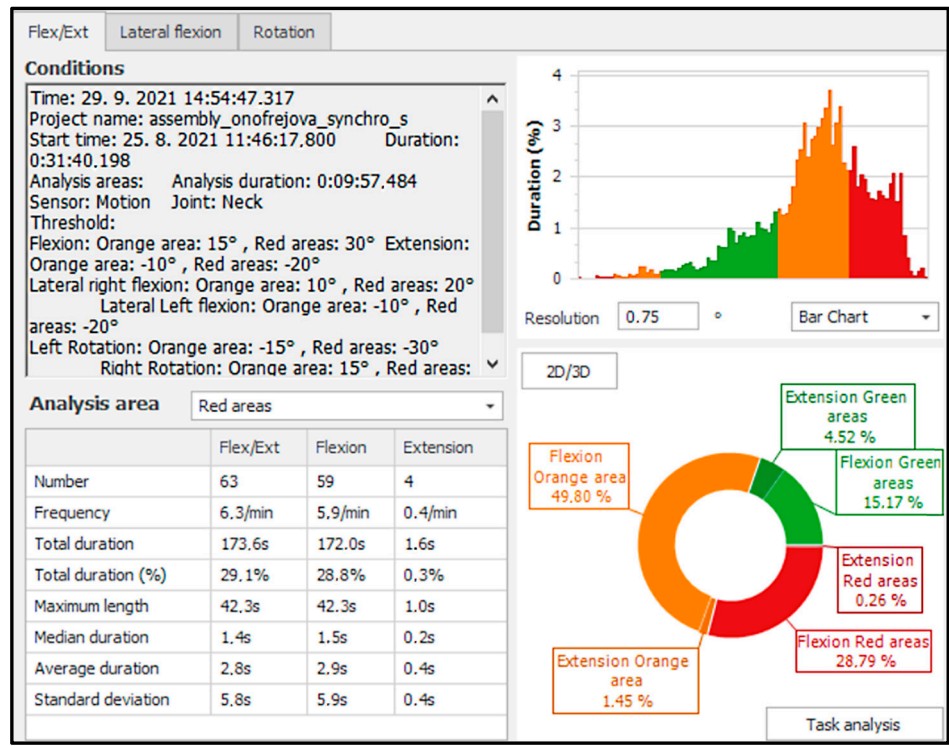

**Figure 4.** Detailed results of time duration in individual postures, in experiments with one worker using exoskeleton: Neck Flexion/Extension.

We provided tests with four different persons. The results were statistically evaluated. The STAND, EXO measured data file normality was verified using the Shapiro-Wilk normality test (Table 4). For each file, we tested the null hypothesis: "The sample distribution is normal." If the *p*-value is less than the significance level $\alpha$, the null hypothesis is rejected, and the distribution is non-normal.

**Table 4.** Verification of measured motion data file normality is performed using the Shapiro-Wilk normality test. For each file we tested the null hypothesis: "The sample distribution is normal." If the *p*-value was less than the significance level $\alpha$, then the null hypothesis was rejected and the distribution was non-normal.

| Joint/Mean $\bar{x}$ [%] | Green Area | | Orange Area | | Red Area | |
|---|---|---|---|---|---|---|
| Condition | STAND | EXO | STAND | EXO | STAND | EXO |
| Neck | 55.5 | 66.7 | 34.1 | 26.7 | 10.4 | 6.5 |
| Lower Back | 93.1 | 95.9 | 6.8 | 4.1 | 0.1 | 0.0 |
| Right Shoulder | 72.8 | 72.0 | 18.6 | 19.0 | 8.6 | 9.0 |
| Left Shoulder | 77.7 [1] | 61.1 [1] | 15.9 [1] | 26.7 [1] | 6.4 | 8.6 |
| Right Hip | 94.6 | 83.7 | 5.4 | 8.0 | 0.0 | 8.2 |
| Left Hip | 89.7 [1] | 72.8 [1] | 10.1 [1] | 21.0 [1] | 0.2 | 6.2 [1] |

[1] Values with significant impact on results.

Because the *p*-value < $\alpha$ for each sample set, we rejected the null hypothesis of normality of the individual base sets. Therefore, we used a nonparametric Wilcoxon pairwise test for pairwise comparisons. The null hypothesis is $H_0$: $m_1 = m_2$, meaning "the medians of both sets are the same," and the alternative hypothesis is $H_1$: $m_1 \neq m_2$, meaning "the medians of both sets are not the same." The resulting *p*-values from pairwise tests are in

Table 5. The null hypothesis was rejected in cases where the *p*-value was lower than the level of significance (*p*-value < α). The test results show statistically significant differences in the case of Left Hip in green and orange areas, Left Shoulder in green and orange areas for all tested subjects (*p*-value < α). It turns out that between the using/eventually not using the exoskeleton in cases Left Hip green, orange, and red areas and Left Shoulder green and orange areas in all tested subjects, there are statistically significant differences. It means that using the exoskeleton brings some ergonomic improvements.

**Table 5.** Test result–paired *t*-test (α = 0.05). Nonparametric Wilcoxon pairwise test for pairwise comparisons with the resulting *p*-values. The null hypothesis (H$_0$: m$_1$ = m$_2$) was "the medians of both sets are the same" and the alternative hypothesis (H$_1$: m$_1$ $\neq$ m$_2$) was "the medians of both sets are not the same."

| Joint/*p*-Value | Green Area | Orange AREA | Red AREA |
| --- | --- | --- | --- |
| Neck | 0.1421 | 0.9061 | 0.3971 |
| Lower Back | 0.4992 | 0.9142 | 0.1187 |
| Right Shoulder | 0.2305 | 0.1198 | 0.4430 |
| Left Shoulder | 0.026 [1] | 0.0010 [1] | 0.5953 |
| Right Hip | 0.2017 | 0.4315 | 0.3265 |
| Left Hip | 0.0092 [1] | 0.0454 [1] | 0.1535 |

[1] Values with significant impact on results.

In summary of all data obtained from the experiments with four workers after statistical evaluation, we obtained the following average results.

The neck seemed to be a critical body segment in the monitored task. The neck's motion was mostly in the green zone (55.5%), except with worker one, but a high time was spent in the red area (10.4%); the worst situation was in neck flexion/extension. The situation was significantly improved while using the exoskeleton; the neck was in the green zone for 66.7%, the orange zone dropped down to 26.7%, and the red from 10.4% to 6.5% (see Table 4).

The lower back was in the optimal green zone almost the entire time (93.1% STAND measurement and 95.9% EXO measurement); the red zone was eliminated when using the exoskeleton (see Table 4).

The right and left shoulders mostly worked in the green zone (more than 70%); however, during work with the exoskeleton, the situation brought higher risk, and the time spent in the orange zone increased for the right shoulder from 18.6% to 19.0%, and for left one from 15.9% to 26.7%. Additionally, significant time was spent in the red zone—right shoulder 8.6% and even with the exoskeleton 9.0%, left shoulder 6.4% and 8.6% with the exoskeleton. The worst situation was identified in horizontal internal/external rotation for both shoulders. The reason for that might be the working table height. When working in the sitting position (EXO), the table still had the same height as during the STAND measurements, breaking the optimal ergonomic conditions.

The right hip was mostly in the green zone (94.6%) but dropped down during the EXO experiment to 83.7%, with some time spent in the red zone (8.2%). Similarly, (Table 4) the left hip was 89.7% in the green zone during STAND and 72.8% during EXO, and 6.2% in the red zone during EXO. The worst situation was in hip internal/external rotation. This might be caused by a limited time for adapting to working with the exoskeleton.

In summary, our experiments justified a positive influence of the CC 2.0 exoskeleton on improvements in ergonomic conditions in the evaluated assembly workplace. As stated in [34,41], over 30% of a shift time in the orange area creates a high risk of developing MSDs. In our case, all segments spent lower time in the orange zone when wearing the exoskeleton.

The Actigraph system offers additional information about workers' load. The average energy spent in 1 h was 40.0 kcal, and the MET Rate (metabolic rate) was 1.66. We used the Freedson Combination (1998) algorithm for energy expenditure assessment and the Freedson Adult (1998) algorithm for metabolic rate assessment. The average heart rate

obtained from the Actigraph smartwatch value reached 72.95 bpm (beats per minute) with SD 3.08, which resulted in moderate activity (the Actigraph distinguishes following bouts of activity: sedentary, light, moderate, vigorous). Normal resting heart rate (RHR) values vary for male adults, generally between 60–70 bpm, and for female adults between 65–80 bpm. We will use a more detailed report from the Actigraph system in connection with further trials in future experiments.

The opinion of the factory workers and ethical recommendations must be a part of an ergonomic analysis looking at the social impact of technologies aiming to improve working conditions like exoskeletons and wearable sensors as stated by authors in [27]. Therefore, we interviewed the workers before and after the experiments. The workers positively accepted the new technology as a potential device aiming to increase their physical performance.

On the other side, workers admit to uncertainty and some discomfort. In our case, positive feedback on the CC 2.0 indicated that three workers accepted the new technology as a technical aid, supporting their physical performance, although they admitted uncertainty when using it in the initial stages. One worker did not feel comfortable with CC 2.0, especially in a phase of short walking when moving to a side part of the working space transferring parts. The worker claimed that sitting was not comfortable, and the cushion often slipped under the backside.

## 4. Discussion

This paper aimed to contribute to the experimental research on MSDs due to their high incidence in industry and the importance of their prevention in the framework of safety and health at work. Regular monitoring of job positions in the industry helps reduce or avoid the risk of injury and occupational diseases and provides comfort and efficient performance at work. In Slovakia, the evidence of WSMDs is still insufficient, and therefore the incidence rate is unknown. Due to the legislation, people with identified disorders have trouble finding a new job after leaving a current position due to the illness and often rehabilitation. Designing healthy workplaces can improve well-being and healthy aging of the population and ambient environment as well as increase efficiency and productivity of operations. As human performance is related to age, sex, muscle strength, body structure, motor skills, the function of sensory organs, and mental ability, some workers might require body support using an exoskeleton, while others in good physical condition might prefer to work without such means. Exoskeletons can become a promising means of overcoming the uneven performance of workers, which often arise from advancing age or gender differences, and thus may become a promising tool provided the right regime of their use, along with the suitability of individual types for specific work activities.

Our strategy for MSDs prevention focuses on evaluating ergonomic risks at the Slovak factories using scientific tools with wearable wireless measurement systems—Captiv and Actigraph. The main issues with our methodology are:

- Performing the inspection of the workplaces for the determination of work activities causing risk using less disturbing measurement methods;
- Exposing the health problems of employees in the musculoskeletal system identified through the modified ergonomic Nordic Questionnaire (NQ-E) [50]. NQ is a symptom questionnaire, designed for all musculoskeletal disorders. The extended version of NQ-E contains some additional questions regarding body postures; job demands and social support;
- Performing a professional assessment of work activities focused on the analysis and prevention of occupational diseases (overuse syndrome, carpal tunnel syndrome, MSDs (musculoskeletal disorders), post-traumatic stress, etc.) due to long-term, excessive, and unilateral load;
- Performing the experiments with and without the exoskeleton in order to verify the suitability and duration of wearing the exoskeletons in Slovak factories.

Researchers such as Mazza et al. [48] and Sabatini [49] raised the question, "Does a sit-stand pattern result in decreasing worker discomfort, injury mechanisms, development of MSDs (especially back disorders), and increasing worker productivity, compared to only standing posture?" Sit-stand work shall be studied in terms of preventing fatigue in the workplace [51,52]. Our contribution is a reaction to that question, and we have already started with research on the sit-stand pattern in the Slovak industrial conditions. However, exoskeletons also offer other functional support for a worker's body, releasing the load on different body parts. Therefore, we will continue in our research to investigate the availability of applying other types of exoskeletons.

Based on the collected anthropometric data used as input for wearable systems Captiv and Actigraph, we are currently able to obtain relevant data for assessing the ergonomic layout of the workplace and take a decision on the adjustment. We are able to compare results about its impact on both the human motion system and human physical behavior from experiments when an occupational exoskeleton is worn or not.

We used the wireless sensor system Captiv for ergonomic risk assessment at the assembly workplace in the automotive industry. Our tested workers repetitively performed assembly of synchronous units in the transmission at a fast pace. Measurement results indicated the unacceptable ergonomic risk in the neck, shoulder, and hip joints. The employee was applied a passive exoskeleton CC 2.0, designed to support the lower limbs to eliminate the physical load. An improvement in the posture is evident in the upper body. Results from our trial measurements show a positive impact on the workers when using the exoskeleton; there is an evident improvement in the position of workers, in flexion of the neck, the ratio (%) between zones green/orange/red was changed from 55.5/34.1/10.4 to 66.7/26.7/6.5; the lower back was without significant changes, the ratio (%) between zones green/orange/red was changed from 93.1/6.8/0.1 to 95.9/4.1/0.0. The greatest improvement was neck flexion/extension. The right shoulder was slightly negatively influenced by the lower position during sitting. The ratio (%) between zones green/orange/red was changed from 72.8/18.6/8.6 to 72.0/19.0/9.0; the worst situation was horizontal internal/external rotation. A similar situation was seen in the left shoulder. Left hip achieved worse results, the ratio (%) between zones green/orange/red was changed from 89.7/10.1/0.2 to 72.8/21.0/6.2, which can be an effect of a short period using a new device—exoskeleton by the employees. The worst situation was observed in the left hip rotation. For the right hip, we observed better conditions than in the left hip, the ratio (%) between zones green/orange/red was changed from 94.6/5.4/0.0 to 83.7/8.0/8.2.

*Limitations*

In our first experiments, we used seven motion sensors due to limited financial sources. It was enough for initial measurements with the following limitations that were necessary to apply: only one joint angle was measured at the arms (left and right upper arms—humerus), and only one joint angle was measured at the upper left and right legs (femur). Therefore, during the evaluation, we concentrated on the upper body and related parts—neck, lower back, right and left shoulder joints, right and left hip joints. We were also able to recognize sitting and standing positions.

In further experiments, we plan to use 15 sensors, adding two on the lower arm at the elbow, two at the wrist joints, two more on the knee joints, and two more on the ankles. The avatar will then copy more complex motions of workers' activities that will give researchers more details for analysis and risk assessment. The optimal number recommended by the Captiv producer company is 15 [47].

Only short training of participants was provided before trial experiments with exoskeletons. Another factor that might have had an impact on results was that the height of experiment participants varied from 1.68–1.85 m.

## 5. Conclusions

WMSDs develop as a consequence of fatigue, arise from limb movements with inappropriate load, and are not particularly harmful in ordinary daily life activities. What makes them hazardous in work situations is the continual repetition, especially at a fast pace, and the lack of time for recovery between them.

This contribution presents our approach to starting a modern monitoring and inspection system for ergonomic improvements, addressing the high incidence of MSDs in Slovak industry, especially in assembly operations in the automotive industry. Ergonomic assessment of physical activity at the workplace provides information on the current state of the physical load of the workers. Simultaneously, we want to consider possible measures, as one of the priority solutions is to test the suitability of exoskeleton implementation, as some factories have already shown interest in such solutions within their ergonomic prevention projects. Exoskeleton manufacturers inform about positive effects mostly based on experiments in the laboratory environment. However, their effect on the industrial environment needs to be verified for a large time frame. The advantage of a multi-sensor system is the collection of complex data at the same time, which simplifies the evaluation and effectiveness of measurements. Knowledge about the influence of exoskeletons on individual parts of the body and the right choice of proper work activities may be beneficial for the design of healthy modern workplaces. It is important to take into account, at the same time, the reflection of persons wearing exoskeletons, including their mental discomfort. By measuring motion using wearable devices (a powerful measurement system) and collecting subjective feedback from workers, we may obtain comprehensive data for assessing the suitability of the exoskeleton for specific work activities.

The advantage of the Captiv sensor system is the collection of complex data simultaneously, which simplifies the evaluation and effectiveness of the measurements. We can identify critical positions of joints, and by visualizations through avatar synchronized with video capture, we can explore which movement caused discomfort. Actigraph smartwatch results add information about human physical activity, especially stress. Such information is important for physically demanding jobs.

In the Slovak Industry, many factories, particularly automotive (VW, Landrover, Kia, PSA Citroën Peugeot,) build their research centers for analyzing the industrial conditions. For this reason, we have chosen the scalable sensor measuring system Captiv, which outputs significant quantification analysis, with the option of fast big data processing, which enables a fast way of suggesting and evaluating measures. Our starting experience with the Captiv system shows that using the system can evaluate big data more precisely and effectively, which can help to improve ergonomic evaluation in the Slovak industry by the scientific approach.

Our ambitions are to provide measurements of human physical behaviour related to the load during work shifts in industry and different workplaces (particularly assembly in the automotive industry) with and without various occupational exoskeletons; passive or active; for the upper body, lower body, or whole body; by wireless multisensor systems such as Captiv and Actigraph. By quantitative measurements, we can carry out long-term observations and acquire valid data to create the methodology of deploying exoskeletons in the Slovak factories.

**Author Contributions:** Conceptualization, D.O.; methodology, D.O. and M.B.; validation, D.O., M.B. and K.V.; formal analysis, Z.K.; investigation, Z.K.; resources, D.O., M.B. and J.G.; data curation, D.O. and K.V.; writing—original draft preparation, D.O.; writing—review and editing, D.O.; visualization, J.G. and D.O.; supervision, K.V.; project administration, D.O. and Z.K.; funding acquisition, Z.K. All authors have read and agreed to the published version of the manuscript.

**Funding:** This research received no external funding.

**Institutional Review Board Statement:** The study was conducted according to the guidelines of the Declaration of Helsinki, and approved by the Institutional Review Board (or Ethics Committee) of Technical University of Kosice (protocol code 8268/2021/R-OLP and date of approval 13 December 2021).

**Informed Consent Statement:** Informed consent was obtained from all subjects involved in the study.

**Data Availability Statement:** Data available on request due to restrictions eg privacy or ethical. The data presented in this study are available on request from the corresponding author. The data are not publicly available due to measurement in private industrial sector.

**Acknowledgments:** This work has been supported by the Slovak Agency Supporting Research and Development APVV-19-0367 and APVV-19-0290 and the Slovak Grant Agency KEGA 013TUKE-4/2020.

**Conflicts of Interest:** The authors declare no conflict of interest. The funders had no role in the design of the study; in the collection, analyses, or interpretation of data; in the writing of the manuscript, or in the decision to publish the results.

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
