# Peer review of "Ergonomic Assessment of Physical Load in Slovak Industry Using Wearable Technologies"

_applsci, doi:10.3390/app12073607_

Round 1
Reviewer 1 Report
This paper presents the ergonomic assessment of physical load where the Chairless Chair 2.0 is used. I find the paper informative and well written. However, the discussion could focus more on the results from the assessment; further, the conclusion section sounds a bit vague in the sense that it does not describe the takeaway sufficiently. Both discussion and conclusion need to be better structured and present the answers to the research question better.
Line 450: The Nordic Questionnaire is not introduced in the paper. Please add a reference. Further, it should be explained what the Nordic Questionnaire is.
Line 461-462: incomprehensible sentence.
Line 434: what do you mean by "question of number of WMSDs"? This is unclear.
On several occasions, "Authors [x]" is used. I thought first that this would be to describe the authors of the paper and have the reference anonymous. However, the same construction is at Line 458. Please consider using the names of the respective paper's authors followed by the citation.
On several occasions, articles are missing. E.g., lines 381, 266, 385, 396, 406, 530, and several more.
Lines 259, 309, and some more: "The results ... represent ..."
Figure 3a: Would it be possible with higher contrast in this figure. Further, it is not readable what the blue text says. Figure 3b: too tiny font. Please enlarge.
Figures 6 and 7: The details in these figures are not readable. Please enlarge.
Line 484: incomprehensible.
Line 486: what does the (%) mean?
Line 499/500: Please provide reference to this claim, and give a hint why this is so.
Lines 511ff: I don't quite understand what this means.
Line 538: Unclear. How can a first experience prove something?
Sections 4 and 5 would need some attention regarding the English language.
As there is a considerable amount of abbreviations and acronyms. Please add a list of the abbreviations and acronyms used in your paper; e.g., MSD, STAND, EXO, WMSD, ...
Reviewer 2 Report
The paper deals with a statistical analysis comparing the gains from using an exoskeleton system.
If the methodology/systematics is the paper's contribution, this information must be clear.
To improve the quality of the paper, authors need to be explicit to the point, concise, and consistent in presenting the problem, methodology, and results.
The paper is long and confusing. A shorter and more consistent text would show the fundamental contributions intended to be presented.
Punctually:
Would Figure 4 be the contribution proposition of the paper?
Figure 6, for example, is difficult to understand, and this one seems cut.
The paper must be reviewed in terms of grammar and formal scientific language. In many points, information is missing.
I make some indications in the attached PDF, but in general, a review of all these above aspects is interesting.
